# Enhancing Structured Evidence Extraction for Fact Verification

**Zirui Wu,  Nan Hu,  Yansong Feng**[*]

Wangxuan Institute of Computer Technology, Peking University, China

{ziruiwu,hunan,fengyansong}@pku.edu.cn

## Abstract

Open-domain fact verification is the task of verifying claims in natural language texts against extracted evidence. FEVEROUS is a benchmark that requires extracting and integrating both unstructured and structured evidence to verify a given claim. Previous models suffer from low recall of structured evidence extraction, i.e., table extraction and cell selection. In this paper, we propose a simple but effective method to enhance the extraction of structured evidence by leveraging the row and column semantics of tables. Our method comprises two components: (i) a coarse-grained table extraction module that selects tables based on rows and columns relevant to the claim and (ii) a fine-grained cell selection graph that combines both formats of evidence and enables multi-hop and numerical reasoning. We evaluate our method on FEVEROUS and achieve an evidence recall of $60.01\%$ on the test set, which is $6.14\%$ higher than the previous state-of-the-art performance. Our results demonstrate that our method can extract tables and select cells effectively, and provide better evidence sets for verdict prediction. Our code is released at https://github.com/WilliamZR/see-st

## 1 Introduction

Open-domain fact verification is the task of verifying a factual claim based on evidence extracted from a knowledge base(Guo et al., 2022; Hardalov et al., 2022). A complete evidence set provides necessary information for a verification module to support or refute the claim. In the real world, fact-checkers could use both unstructured and structured data as evidence.

Tables are a structured form of evidence that provides an organized representation of information that facilitates effortless comparison and cross-referencing. By presenting information in cells,

---

[*] Corresponding Author.

Figure 1: Example of FEVEROUS with gold evidence. The gold cell evidence is highlighted in orange.

tables enable efficient look-up of individual values, while the arrangement of rows and columns allows for the extraction of high-order semantics through various operations, such as comparisons, filtering, arithmetic calculations, and the identification of minimum and maximum values. The two-dimension structure and high-order semantics of tables make them more difficult to extract compared with texts. Regarding the previous method to extract structured evidence. Aly et al. (2021) treats tables as text sequences neglecting the table structure itself and extracts them only based on lexical matching. This approach is also used in many following works (Bouziane et al., 2021; Saeed et al., 2021; Hu et al., 2022, 2023). Consequently, these methods struggle to extract tables with little word overlap with the claim. Alternatively, Bouziane et al. (2021) and Gi et al. (2021) convert each table cell into a text sequence and perform cell retrieval directly. Kotonya et al. (2021) construct a reason-

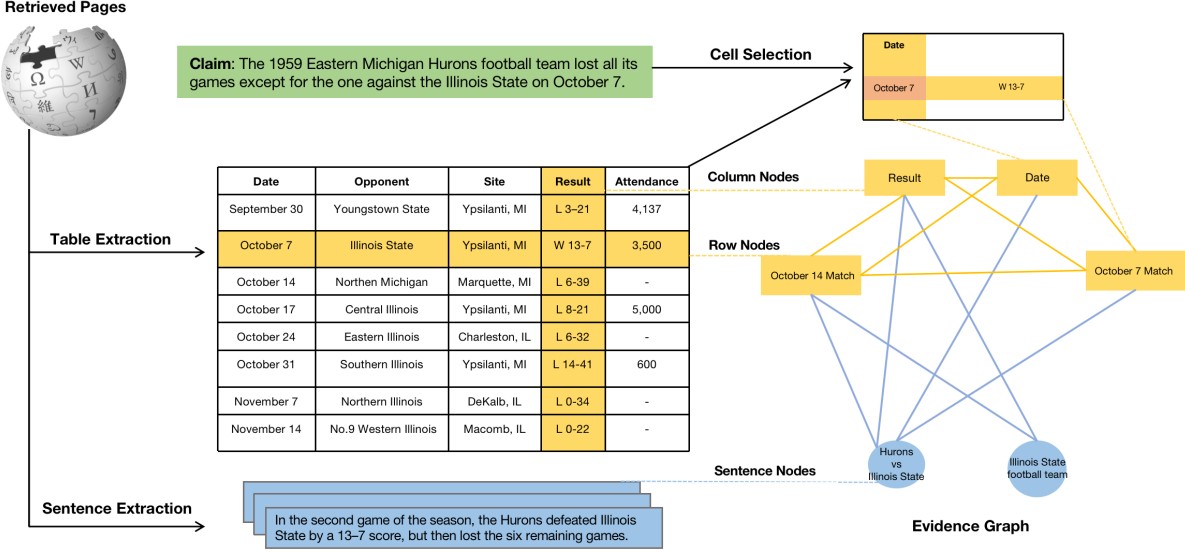

Figure 2: Overview of our proposed approach SEE-ST. It extracts structured evidence based on retrieved pages and extracted sentences. SEE-ST includes two modules: coarse-grained table extraction (§2.1) and fined-grained cell selection based on an evidence graph(§2.2).

ing graph of sentences and cells to extract evidence. Yet these methods overlook the context within the table and fail to capture high-order semantics from rows and columns.

Figure 1 presents an example from FEVER-OUS (Aly et al., 2021), a benchmark on open-domain fact verification using both sentences and table cells as evidence. The verification model is expected to extract both unstructured and structured evidence to verify the claim. The structured table can be interpreted in several ways for the model: a complete table, a combination of rows or columns, or a composition of isolated cells. The entire table encompasses a substantial amount of information. During table extraction, the *Site* and *Attendance* columns are irrelevant and only introduce confusion for both lexical and semantic methods (Herzig et al., 2021). Cells typically contain short phrases and numbers, which are often insufficient for conveying comprehensive information. In contrast, rows and columns possess a similar volume of information to the claim, thereby exhibiting comparable semantic granularity. The extraction process should prioritize the information from the 3rd row and the *Result* column. During the cell selection step, the selection module must capture the high-order semantics in the *Result* column to verify the statement ... *lost all its games except for one....* Moreover, it needs to identify the cells *October 7* and *Illinois State* by integrating informa-

tion from the *Date* and *Opponent* columns and the 3rd row. Overall rows and columns have comparable semantic granularity with the claim and are the most suitable fundamental unit for both table extraction and cell selection.

In this paper, we propose **S**tructured **E**vidence **E**xtraction with Row and Column **S**emantics of **T**ables (SEE-ST), a novel approach to enhance both table extraction and cell selection modules for fact verification by leveraging the semantics of rows and columns. This method aligns the semantic granularity of structured evidence with claim and sentence evidence, enabling more precise evidence extraction. SEE-ST starts by extracting tables from retrieved Wikipedia pages and focuses on identifying the most relevant rows and columns for the claim, thereby reducing confusion arising from irrelevant cells. Since a cell represents the intersection of a row and a column, its selection can be conducted by analyzing both dimensions. We integrate rows and columns within a graph neural network to facilitate fine-grained cell selection while preserving high-order semantics. Additionally, we incorporate extracted sentence evidence into the graph to merge information from evidence of both formats at a similar semantic granularity.

Our contributions can be summarized as follows: (i) We propose to leverage the row and column semantics to enhance table extraction and cell selection. (ii) We design an evidence graph for cell

selection. It enables rows and columns to interact with sentences at comparable semantic granularity while still maintaining high-order semantics from table evidence. (iii) Our proposed method SEE-ST achieves state-of-the-art performance in evidence extraction on the FEVEROUS dataset, with better evidence sets contributing to the increased accuracy of the verification module.

## 2 Our Model

FEVEROUS is a benchmark on open-domain fact verification (Aly et al., 2021). The task is to verify a claim $c$ based on Wikipedia. We follow the three-step pipeline in open-domain fact verification (Thorne et al., 2018; Aly et al., 2021). The three steps are document retrieval, evidence extraction, and verdict prediction. The first step retrieves $n_p$ relevant pages from the Wikipedia dump. Then the second step extracts sentence evidence $S$ and cell evidence $C$ from the retrieved pages. At last, the model predicts the veracity label $\hat{y}$ of the claim based on sentence and cell evidence.

SEE-ST enhances the extraction of structured evidence in the second step of the pipeline. It operates on the same conceptual framework, utilizing row and column semantics to enhance evidence extraction (Figure 2). SEE-ST performs structured evidence extraction in two stages: (i) Coarse-grained table extraction that retrieves top $n_t$ tables for the second step. It scores relevance between rows or columns with the claim and ranks tables accordingly and (ii) Fine-grained cell selection that retrieves cell evidence from complex tables. It constructs a graph neural network of rows, columns, and sentences at a comparable semantic granularity. It utilizes a graph attention network to enable delicate information integration between different pieces of evidence. The cells are selected as the intersection of rows and columns.

At the verification step, we utilize DCUF (Hu et al., 2022), a method that converts evidence into dual-channel encodings to verify the claim.

### 2.1 Coarse-grained Table Extraction

In this section, we introduce our table extraction module. First, the claim and table pair are fed to TAPAS, a pre-trained table model aware of table structures(Herzig et al., 2020). Instead of using embedding of [CLS] token to predict the relevance between the claim and table. our module predicts the relevance based on row and column semantics.

Each row and column in the evidence table is represented as the average of all the token embeddings in the same row or column. Then we predict a probability score for each row and column separately.

The selective score for the table is computed as the multiplication of the maximum probability score for the rows and columns.

$$\mathrm{P}(t) = \max\left(\mathrm{P}_{row}\right) \times \max\left(\mathrm{P}_{col}\right)$$

The training set is constructed with all the tables that contain gold cell evidence as positive examples. As for negative examples, we use top tables retrieved by DrQA (Chen et al., 2017) which do not contain cell evidence but have lexical overlap with the claim. The ratio of positive and negative examples is 1 at every mini-batch in training. During training, the model is trained to select table rows and columns that contain gold cell evidence. The loss function is computed with a cross-entropy loss on each row and column.

$$L = \alpha_t L^{Row} + \beta_t L^{Col}$$

$L^{Row}$, and $L^{Col}$ are cross-entropy loss functions of rows and columns. These rows and columns that have gold cell evidence are labeled as *select* while rows/columns in the same table but do not contain cell evidence and rows/columns in negative examples are labeled as *not select*.

At inference, a selective score is computed for each table in retrieved pages. Top $n_t$ tables are selected for the following fine-grained cell selection.

### 2.2 Fine-grained Cell Selection

Following the idea of leveraging row and column semantics, we build an evidence graph $\mathcal{G}(\mathcal{V}, \mathcal{E})$ to select cell evidence. The first two types of nodes $v_0, ..., v_M$ and $v_{M+1}, ..., v_{M+N}$ encode table rows and columns. Shown as yellow nodes in Figure 2. $M$ and $N$ are the total number of table rows and columns. The third type of node is sentences $v_{M+N}, ..., v_K$, shown as blue nodes in Figure 2. Overall, the evidence graph has $K$ nodes of unstructured and structured evidence at a comparable semantic granularity.

Graph edges $\mathcal{E}$ are constructed between related evidence following three rules: (i) Cross-Format Related: $v_i$ and $v_j$ are different format evidence of sentence and table, and in the same Wikipedia page. (ii) Structure-Based: $v_i$ and $v_j$ are nodes from the same table and (iii) Entity-Based: $v_i$ and $v_j$ have

common entities and hyperlinks or sentences on the same Wikipedia page.

We feed each claim-sentence pair to a RoBERTa model and use the embedding for [CLS] token to initialize the sentence nodes in our graph. And nodes of table rows and columns are initialized using the same method as in §2.1.

Then we apply a layer of Graph Attention Network (GAT) (Veličković et al., 2018) with a residual connection to update representations of each node. It dynamically allocates weights between different nodes of evidence, allowing the graph to capture semantic connections between evidence.

Finally, we use a two-layer feed-forward network to predict the retrieved probability for each node separately. The selective score for cell evidence is the multiplication of the probability of the row node and column node that contains the cell. Top $n_c$ cells are extracted for verdict prediction.

The loss function is computed as the weighted sum of the loss on nodes and cells:

$$L = L^S + \alpha_c \cdot L^{Col} + \beta_c \cdot L^{Row} + \gamma_c \cdot L^{Ce}$$

$L^S$, $L^{Col}$, $L^{Row}$, and $L^{Ce}$ are cross-entropy loss computed on sentences, table columns, table rows, and cells. $\alpha_c$, $\beta_c$ and $\gamma_c$ are coefficients to adjust weights between nodes and cells. Nodes of sentences in the gold evidence set are labeled as positive. Nodes of rows and columns are labeled as positive if they contain the gold cell evidence.

## 3 Experiments

We utilize FEVEROUS as the test bed for our approach, as it is, to our knowledge, the only open-domain fact verification benchmark that combines both unstructured and structured evidence. FEVEROUS aims to accomplish two objectives: first, to extract sentence and table cell evidence from English Wikipedia, which contains over 95.6 million sentences and 11.8 million tables; and second, to predict the veracity label $\hat{y}$ of a given claim. The possible veracity labels include SUPPORTS, REFUTES, and NOT ENOUGH INFO (NEI). Some claims can be verified using either sentences or tables alone, while approximately 59% of instances need structured evidence for claim verification. 27% of the instances need to combine evidence from unstructured and structured evidence to verify. The quantitative characteristics of the benchmark are presented in Table 1. Detailed experiment settings can be found in Appendix A.

|  | Train | Dev | Test | Total |
|---|---|---|---|---|
| Supported | 41,835 | 3,908 | 3,372 | 49,115 |
| Refuted | 27,215 | 3,481 | 2,973 | 33,669 |
| NEI | 2,241 | 501 | 1,500 | 4,242 |
| Sentences | 31,607 | 3,745 | 3,589 | 38,941 |
| Cells | 25,020 | 2,738 | 2,816 | 30,574 |
| Sentences+Cells | 20,865 | 2,468 | 2,062 | 25,395 |
| Total | 71,291 | 7,890 | 7,845 | 87,026 |

Table 1: Quantitative characteristics of FEVEROUS.

### 3.1 Evaluation

Evidence extraction is crucial in fact verification, it provides the verdict prediction module with the necessary evidence to verify the claim. FEVEROUS uses precision, recall and F1 score to measure the quality of retrieved evidence. $\mathbb{E}$ is the collection of gold evidence sets and $\hat{E}$ is the retrieved evidence set. Recall of a specific type of evidence is computed for an instance as:

$$\text{Recall}(\mathbb{E}, \hat{E}) = \frac{|\hat{E} \cap \cup \mathbb{E}|_i}{|\cup \mathbb{E}|_i}$$

$|E|_i$ is the number of $i$ type evidence in set $E$. Evidence type $i$ could be sentence, table or cell.

A retrieved evidence set is considered complete iff at least one gold evidence set is included. The recall of evidence set is defined for an instance as follows:

$$\text{Recall}(\mathbb{E}, \hat{E}) = \begin{cases} 1 & \exists E \in \mathbb{E} : E \subseteq \hat{E} \\ 0 & \text{otherwise} \end{cases}$$

The number of retrieved evidence is limited to 5 sentences and 25 cells at maximum (Aly et al., 2021). In the following discussion, recall refers to evidence recall unless specified for an evidence type.

The evaluation for verdict prediction uses label accuracy(Acc.) and FEVEROUS score(F.S) as its metrics. They are defined for an instance as:

$$\text{Acc.}(y, \hat{y}) = \begin{cases} 1 & y = \hat{y} \\ 0 & \text{otherwise} \end{cases}$$

$$\text{F.S} = \text{Acc.}(y, \hat{y}) \times \text{Recall}(\mathbb{E}, \hat{E})$$

$y$ is the gold label and $\hat{y}$ is the predicted label. FEVEROUS score considers both the correct prediction of the veracity label and the completeness of the retrieved evidence set.

| Models | Table | Sentence | Cell | Evidence |
|--------|-------|----------|------|----------|
| Baseline | 56 | 53 | 29 | 30 |
| FaBULOUS | - | 56.6 | 34.2 | 40.4 |
| DCUF | 75.59 | 62.54 | 58.41 | 43.22 |
| UnifEE | 75.59 | 75.36 | 67.44 | 55.08 |
| Our Model | **80.86** | 75.50 | **77.16** | **61.43** |

Table 2: Recall of different formats of evidence on the development set.

## 3.2 Main Results

**Evidence Extraction Results** Table2 shows the extraction results of our model on the development set for different categories of evidence. All the previous methods restrict their evidence set to at most 5 sentences, 3 tables, and 25 cells. Official baseline (Aly et al., 2021) and FaBULOUS (Bouziane et al., 2021) use a weaker document retrieval module which results in error propagation and low evidence recall. Our model uses the same document retrieval module as DCUF (Hu et al., 2022) and UnifEE (Hu et al., 2023), which obtains a recall of $85.20\%$ for Wikipedia pages. Therefore recall of each format of evidence can be used to compare the performance of evidence extraction directly. SEE-ST leverages the row and column semantics of tables and increases the recall of tables by 5.27%. SEE-ST also increases the recall of cells by 11.65% and 9.27% compared with DCUF and UnifEE. This suggests the two stages of SEE-ST, table extraction and cell selection can both improve the performance of structured evidence extraction.

**Overall Results** Table 3 shows overall performance on the dev and test set based on our evidence extraction results. We achieve 60.01% evidence recall on the blind test set, which is 6.14% higher than the previous state-of-the-art UnifEE (Hu et al., 2023). Following the verification method proposed in Hu et al. (2022), we obtain accuracy 74.68%/65.16% and FEVEROUS score 49.73%/44.75% on dev/test set. This demonstrates our enhancement on structured evidence extraction provides verification with more accurate evidence so that it could make the right predictions.

Since SEE-ST obtains the maximum number of evidence in extraction, we also experiment on UnifEE without its threshold selection mechanism to compare its performance with SEE-ST more fairly. SEE-ST still improves the evidence recall by a large margin. Our choice of discarding threshold selection is further analyzed in Appendix B.

The accuracy on the test set is generally about 10% lower than the accuracy on dev set. The main reason is the unbalanced instances of NEI claims in different splits. Our analysis of verdict prediction results shows that DCUF performs poorly in NEI instances. This explains the accuracy gap between the dev and test set.

## 3.3 Ablation Study

**Table Extraction** We assess the effectiveness of our table extraction model through the following ablation experiments: (i) w/o Column Semantics: We train a model that only considers the relevance between rows and the claim for table extraction, scoring tables based on the criterion $\max{(\mathrm{P}_{row})}$. (ii) w/o Row Semantics: Analogously, we score tables using $\max{(\mathrm{P}_{col})}$. (iii) w/o Row and Column: We eliminate the design of table extraction and instead utilize the embedding of the [CLS] token in claim-table encodings for table extraction. The recall of the top 3 tables is presented in Table 4.

In the setups without column or row semantics, the recall of the top 3 tables decreases by 1.03% and 1.76%, respectively. In the w/o Row and Column setting, table recall drops significantly by 17.75%, resulting in a recall of 63.11%, which is even lower than the recall achieved by DrQA (Table 2). These results suggest that our table extraction module in SEE-ST, which combines semantics from both rows and columns, effectively captures the coarse-grained relevance between claims and tables.

**Cell Selection** To evaluate the effectiveness of our evidence graph for cell selection, we design ablation experiments with an evidence graph constructed with the top 3 tables. The experimental settings are as follows: (i) w/o Structured Edges: We introduce additional edges between rows and columns from the same Wikipedia pages but not the same table. (ii) w/o Edge Pattern: We directly connect every node in the evidence graph to each other. (iii) w/o Row Semantics: Following Hu et al. (2023), we build an evidence graph of sentences and columns, and separately, a graph of table cells, using columns as intermediates to select cells. (iv) w/o Evidence Graph: We follow the table extraction pipeline with an added term for the cell loss function, selecting each cell based solely on the context within the same table.

The results of the ablation experiments are presented in Table 5. When increasing the number of tables in the evidence graph from 3 to 5, cell and

| Models | Dev | | | | | Test | | | | |
|---|---|---|---|---|---|---|---|---|---|---|
| | F.S | Acc. | E-P | E-R | E-F1 | F.S | Acc. | E-P | E-R | E-F1 |
| Official Baseline | 19 | 53 | 12 | 30 | 17 | 17.73 | 48.48 | 10.17 | 28.78 | 15.03 |
| EURECOM | 19 | 53 | 12 | 29 | 17 | 20.01 | 47.79 | 13.73 | 33.73 | 19.52 |
| Z team | – | – | – | – | – | 22.51 | 49.01 | 7.76 | 42.64 | 13.12 |
| CARE | 26 | 63 | 7 | 37 | 12 | 23 | 53 | 7 | 37 | 11 |
| NCU | 29 | 60 | 10 | 42 | 17 | 25.14 | 52.29 | 9.91 | 39.07 | 15.81 |
| Papelo | 28 | 66 | – | – | – | 25.92 | 57.57 | 7.16 | 34.60 | 11.87 |
| FaBULOUS | 30 | 65 | 8 | 43 | 14 | 27.01 | 56.07 | 7.73 | 42.58 | 13.08 |
| DCUF | 35.77 | 72.91 | 15.06 | 43.22 | 22.34 | 33.97 | 63.21 | 14.79 | 44.10 | 22.15 |
| UnifEE | 44.86 | 73.67 | 19.04 | 55.08 | 28.30 | 41.50 | 65.04 | 18.35 | 53.87 | 27.37 |
| UnifEE* | 46.13 | 73.14 | 12.48 | 56.22 | 20.42 | - | - | - | - | - |
| Our Model | **49.73** | **74.68** | 10.60 | **61.43** | 18.07 | **44.75** | **65.16** | 9.81 | **60.01** | 16.89 |

Table 3: Model performance on the development set and test set. F.S is FEVEROUS score and Acc. is the accuracy of veracity labels. E-R, E-P and E-F1 are recall, precision and F1 computed based on the evidence set. All metrics are averaged on all instances. UnifEE* uses the maximum number of evidence as the constraint for the evidence set.

| Models | Recall@3 |
|---|---|
| SEE-ST | **80.86** |
|   w/o Column Semantics | 79.83 |
|   w/o Row Semantics | 79.10 |
|   w/o Row and Column | 63.11 |

Table 4: Ablation study of coarse-grained table extraction. Recall@3 is the recall of top 3 extracted tables.

| Recall | Cell | Evidence |
|---|---|---|
| Top5 Tables | | |
|   SEE-ST | **77.16** | **61.43** |
| Top3 Tables | | |
|   SEE-ST | 74.20 | 60.16 |
|     w/o Structured Edges | 72.30 | 58.61 |
|     w/o Edge Pattern | 72.22 | 58.56 |
|     w/o Row Semantics | 73.16 | 59.24 |
|     w/o Evidence Graph | 68.37 | 56.08 |

Table 5: Ablation study of fine-grained cell selection.

evidence recall rise by 2.96% and 1.38%, respectively, achieving a recall rate of 77.16% for cell evidence and 61.43% for evidence recall. In the w/o Structured Evidence setting, rows and columns from different tables are connected if they are on the same Wikipedia page, resulting in a decrease of 1.98% in cell recall and 1.53% in evidence recall. This demonstrates that the Structured Edges designed in §2.2 are beneficial for selecting cells as intersections of rows and columns. Connecting all cells in the graph further compromises the performance of structured evidence. In the w/o Row Semantics setting, we build two evidence graphs following Hu et al. (2023): one containing sentences

and columns, and another with cells. Cell recall and evidence recall decline by 1.04% and 0.92% in this setting, indicating that SEE-ST's utilization of row and column representations captures cell information more precisely. Moreover, compared to the two graphs of sentences and cells, SEE-ST is more computationally efficient with only one evidence graph. Finally, the w/o Evidence Graph setting loses all connections between sentences and tables, as well as shared entity information among evidence. Each node is predicted based on its own information, resulting in a significant drop in cell and evidence recall. This highlights the importance of the design of evidence graph that enables interactions between texts and tables.

## 4 Analysis

### 4.1 Error Analysis

Here we perform a detailed error analysis for evidence extraction.

**Error Source Analysis** The error source for evidence sets is defined as the format of evidence that is not extracted completely in the three-step pipeline (§2). We use *Page* to denote that document retrieval fails to retrieve all the pages containing evidence. For those instances that have a complete document set, the error source is defined according to the format of evidence fail to extract: *Unstructured* (sentences), *Structured* (tables or cells) and *Both*. The instances with a complete evidence set are denoted as *Complete*. About 1% of instances have other evidence types such as lists and table captions. These instances are excluded during error analysis. The impact of ignorance of other evidence types is discussed in the Limitations section.

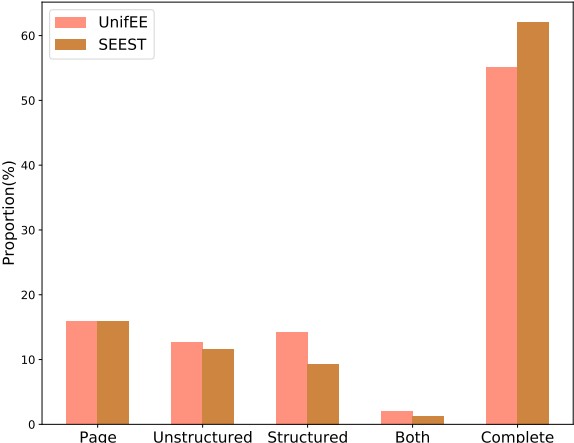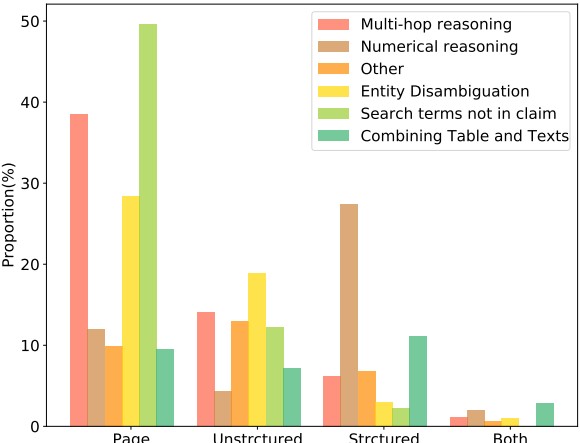

Figure 3: Error analysis of extracted evidence set for the dev set. Left: Overall error source analysis. Right: Error source proportions of claims with different reasoning challenges.

The result of the error source analysis is shown in Figure 3. The document retriever fails to retrieve complete documents for 15.8% instances in the dev set. Our proposed method decreases the proportion of cases missing structured evidence from 14.28% to 9.34%. It shows the effectiveness of SEE-ST in structured evidence extraction.

**Analysis based on challenge types** A fact-checker system would face a specific reasoning challenge when extracting evidence and verifying claims for FEVEROUS. The challenges include *multi-hop reasoning*, *entity disambiguation*, *search term not in claim*, *combining tables and texts* and *numerical reasoning*. The challenges that do not belong to those five types are categorized as *other*.

We further evaluate the performance of SEE-ST in each category to demonstrate its ability to retrieve evidence for claims with different reasoning challenges. SEE-ST achieves great extraction performance over claims that need to combine tables and texts with a recall rate of 69.27% showing that the unified graph neural network can effectively integrate information from both formats of evidence. The recall rates of claims that need multi-hop reasoning, entity disambiguation or search terms are lower than 50%. But the analysis in Figure 3 shows that the main error source of incomplete evidence extraction is document retrieval. It fails to retrieve complete document sets for each challenge type listed above with an error rate of 38.6%, 28.4% and 54.6% while only about 7% instances in these challenges lose complete evidence sets because of unsuccessful structured evidence extraction. This demonstrates the ability of our model to retrieve evidence in more challenging scenarios. Our model

also improves the evidence extraction performance of numerical reasoning. The rate of *Structured* errors is 34.82% for UnifEE (Hu et al., 2023). SEE-ST increases the recall rate by 8.25% and decreases the rate of *Structured errors* by 7.35%. Yet numerical reasoning remains the main challenge in structured evidence extraction.

## 5 Case Study

In this section, we demonstrate two cases for evidence extraction in Figure 4. For the claim on *1991 Waterford City Council election*, our table extraction module successfully extracts the table of election statistics from the Wikipedia dump. The main challenge for this case is numerical reasoning. Verifying ... *six parties all of which have at least two seats* requires high-order semantics in *Seats* column. Such semantics is maintained in coarse-grained table extraction and fine-grained cell selection. SEE-ST extracts cells in *Party* and *Seats* columns as evidence. It shows the numerical reasoning ability of SEE-ST.

As for the evidence extraction for the claim of *The Castle of Iron*, it demonstrates the ability of SEE-ST to perform multi-hop reasoning. The claim states that ...*was published by a large-scale publishing company in New York*, and is refuted by sentence evidence *Gnome Press was an American small-press publishing company...*. However, the name of the publisher is not mentioned in the claim. It is the structured evidence that links the claim with the sentence evidence. The design of our evidence graph could enable interaction between sentence and table cells to achieve multi-hop reasoning over different formats of evidence.

<table>
<tr><td colspan="2">

**Claim**: 1991 Waterford City Council election was planed to take place on 27 June 1991 with six parties all of which have at least two seats.

**Label**: SUPPORTS      **Challenge**: Numberical Reasoning

</td></tr>
</table>

| **Claim**: 1991 Waterford City Council election was planed to take place on 27 June 1991 with six parties all of which have at least two seats. | **Claim**: The Castle of Iron is a fantasy novel written by L. Sprague de Camp and Fletcher Pratt and was published by a large-scale publishing company in New York. |
|---|---|
| **Label**: SUPPORTS    **Challenge**: Numberical Reasoning | **Label**: REFUTES    **Challenge**: Multi-hop Reasoning |

**Extracted Evidence**

**Structured:**

| Party | Seats | ± | First Pref. Votes | FPv% |
|---|---|---|---|---|
| Fianna Fáil | 3 | -2 | 3,165 | 19.01% |
| Fine Gael | 2 | -2 | 2,220 | 13.34% |
| Labour | 3 | +1 | 3,415 | 20.5% |
| Workers's Party | 3 | +1 | 3,359 | 20.18% |
| Progressive Democrats | 2 | +2 | 1,666 | 10.01% |
| Independent | 2 | - | 2,360 | 14.2% |
| Totals | 15 | - | 16,645 | 100% |

...

**Untructured:**

An election to Waterford City Council took place on 27 June 1991 as part of that year's Irish local elections.
Waterford City Council Comhairle Cathrach Phort Láirge) was the authority responsible for local governments in the city of waterford.
...

**Extracted Evidence**

**Structured:**

| Author | L. Sprague de Camp and Fletcher Pratt | Publisher | Gnome Press |
|---|---|---|---|
| Cover artist | Hannes Bok | Publication date | 1941, 1950 |
| Country | United States | Media type | Print |
| Language | English | Pages | 224 |
| Series | Harold Shea | Preceded by | The Incomplete Enchanter |
| Genre | Fantasy | Followed by | Wall of Serpents |

...

**Untructured:**

Gnome Press was an American small-press publishing company primarily known for publishing many science fiction classics.
The Castle of Iron is the title of a fantasy novella by American authors L. Sprague de Camp and Fletcher Pratt, and of the novel into which it was later expanded by the same authors.
...

Figure 4: Demonstrations of extracted evidence set for claims in FEVEROUS. Gold evidence is highlighted in orange. The underlines indicate the cells selected by SEE-ST and table headers are marked with bold texts.

## 6 Related Work

Open-domain fact verification extracts evidence from a knowledge base (KB) to verify the claim. In the real world, unstructured and structured evidence can be used to verify the claim. FEVER (Thorne et al., 2018) and HoVer (Jiang et al., 2020) focus on extracting unstructured evidence i.e. sentences from the Wikipedia dump. And the research of structured evidence mainly focuses on tables. TabFact (Chen et al., 2019) and InfoTabs (Gupta et al., 2020) verify claims with tables and infoboxes from Wikipedia. SEM-TAB-FACT (Wang et al., 2021) and SciTab (Lu et al., 2023) are constructed on claims and tables from scientific papers. Yet these datasets all provide the table evidence for verification as in a closed-domain setting. To our knowledge, FEVEROUS is the only open-domain fact verification dataset that considers table as evidence (Aly et al., 2021). Therefore, our work uses FEVEROUS as a testbed for table extraction and cell selection.

As for table extraction, the lexical matching approach is widely used in fact verification (Aly et al., 2021; Hu et al., 2022, 2023). Chen et al. (2020) encodes tables with BERT and concatenate embedding with structured features to rank tables. TAPAS enhances table encoding with additional table-aware positional embeddings (Herzig et al., 2021). (Pan et al., 2021) also leverages represen-

tations of row and column to rank tables but overlooks the context of other cells. TaBERT (Yin et al., 2020) utilizes a content-snapshot mechanism to only encode the relevant rows thus may lose the high-order semantics of columns.

Gi et al. (2021) converts each cell into a text sequence to select cell evidence, bypassing the table extraction step. Acharya (2021) selects cells through dependency parsing. Jindal et al. (2021) combines single-cell NLI task with cell-wise relevance to extract evidence at cell-level semantic granularity. These methods are performed at the semantic granularity of cells. Approaches in question answering have proposed to select cells as the intersection of rows and columns (Glass et al., 2021; Pan et al., 2021). All these above methods do not consider the interaction between evidence.

The idea of evidence graph has been explored in previous works. Kotonya et al. (2021) linearizes cells into sentences to construct a fully connected evidence graph with texts. UnifEE (Hu et al., 2023) proposes to use column nodes as intermediates between sentence-level and cell-level evidence graphs. Our method is more computationally efficient with only one evidence graph. Our design enables rows and columns to interact with sentences at comparable semantic granularity while still maintaining high-order semantics from tables.

# 7 Conclusions

This paper presents SEE-ST, a novel method for enhancing the extraction of structured evidence. By leveraging both row and column semantics, SEE-ST can operate with a comparable evidence granularity, enabling both coarse-grained table extraction and fine-grained cell selection. Experimental results demonstrate the effectiveness of SEE-ST, as it significantly improves the recall of structured evidence compared to existing methods in fact verification. Moreover, SEE-ST outperforms state-of-the-art methods in evidence recall and FEVEROUS score on FEVEROUS benchmark. These findings underline the impact of SEE-ST in enhancing structured evidence extraction.

## Limitations

Our work explores the extraction of structured evidence in enhancing table extraction and cell selection. In the real world, fact checkers also use other structured evidence as knowledge graphs and databases to verify the claim. FactKG is a fact verification dataset on knowledge graphs (Kim et al., 2023). The verification model is required to retrieve subgraphs as evidence for verdict prediction. SEE-ST can not be directly applied to this task. We will try to build a unified evidence extraction module that could retrieve both tables and subgraphs for fact verification in our future work.

Our work only enhances the extraction of structured evidence, while the whole evidence extraction process still suffers from error propagation from document retrieval. The error propagation in the three-step pipeline makes it extremely difficult for the model to retrieve complete evidence sets and predict the right verdict label. A new pipeline for open-domain fact verification is needed to solve this issue.

## Acknowledgements

This work is supported by NSFC (62161160339). We would like to thank the anonymous reviewers for their helpful comments and suggestions. For any correspondence, please contact Yansong Feng.

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

## A  Implementation Details

In our approach, we retrieve $n_p = 5$ pages from Wikipedia for each claim following Hu et al. (2022). And we extract $n_s = 5$ sentences and $n_t = 5$ tables

from the pages to construct our evidence graph. As for the cell selection, we limit the selected cells to a maximum of $n_c = 25$ cells. For a fair comparison with baselines, the source tables of cells are constrained to the top 3 tables. The tables are re-ranked by the highest cell selective score at cell selection.

For table extraction, we utilize TAPAS-base[1] to encode the claim-table pair. Our model is trained with Adam optimizer (Kingma and Ba, 2014) with a batch size of 8. The learning rate is $10^{-7}$ for TAPAS and $10^{-5}$ for the classifier. $\alpha_t$ and $\beta_t$ are assigned a value of 1.

Regarding our evidence graph, we employ Spacy[2] to extract common entities from evidence to connect relevant nodes. RoBERTa-base[3] and TAPAS-base are chosen as the encoders for node initialization. The model is optimized using the Adam optimizer (Kingma and Ba, 2014) with a batch size of 4 and a peak learning rate of $10^{-6}$. We set the weights for different coefficients of loss functions as follows: $\alpha_c = \beta_c = 2$ and $\gamma_c = 1$. The model takes 14 hours to train for 3 epochs and 34 minutes to select cells in the dev set on a single NVIDIA A100 GPU.

## B  Evidence Precision Analysis

With the constraint of a maximum number of sentences and cells in the evidence set, there are two ways to construct evidence sets. The previous SOTA method UnifEE (Hu et al., 2023) uses a threshold to construct an evidence set with high precision at the expense of a slight decrease in evidence recall. Other baselines use evidence as much as fit the constraints to construct the evidence sets resulting in higher evidence recall. In this section, we aim to analyze which strategy achieves higher accuracy and FEVEROUS score at the verdict prediction step. We filter out unnecessary evidence with different thresholds and verify claims based on the filtered evidence sets. The results are presented in Table 6.

The results indicate that as the evidence threshold increases, evidence precision improves at the cost of reduced evidence recall. However, label accuracy remains largely unaffected by changes in the evidence set. Threshold selection leads to a decrease in the FEVEROUS score. Overall, the

[1] https://huggingface.co/google/tapas-base
[2] https://spacy.io/
[3] https://huggingface.co/roberta-base

verification model used in this work is robust to threshold selection within the same extracted evidence set, and verdict prediction achieves the highest FEVEROUS score without threshold selection. The annotations of the gold evidence set may not be exhaustive of all possible evidence, which could result in an underestimation of precision due to the presence of correct evidence that is not annotated as gold evidence. In summary, SEE-ST employs the maximum number of sentences and cells as constrain for evidence sets for optimal verdict prediction performance.

| Threshold | E-P | E-R | E-F1 | F.S | Acc. |
|---|---|---|---|---|---|
| 0 | 10.60 | 61.43 | 18.07 | **49.73** | **74.68** |
| 0.001 | 12.99 | 60.17 | 21.36 | 48.38 | 74.25 |
| 0.005 | 15.85 | 59.60 | 25.04 | 48.04 | 74.03 |
| 0.01 | 17.88 | 58.54 | 27.49 | 47.28 | 74.22 |
| 0.05 | 23.39 | 50.78 | 32.04 | 42.56 | 73.99 |

Table 6: Effects of evidence extraction on verdict prediction.

| Verdict | Recall |
|---|---|
| NEI | 41.72 |
| SUPPORTS | 61.38 |
| REFUTES | 61.84 |

Table 7: The performance of evidence extraction for claims with different veracity labels.

## C  Performance on Different Verdict

We further analyze the performance of evidence extraction and verdict prediction for claims with different veracity labels, as shown in Table 7. Generally, extracting evidence for *REFUTES* claims is more challenging, as the information contained in the evidence is not always explicitly included in the claim, making it difficult to retrieve based on relevance alone. However, our model achieves comparable extraction performance for both *SUPPORTS* and *REFUTES* types, with recall rates of 61.38% and 61.84%, respectively. The recall for *NEI* instances is approximately 20% lower than for the other types. NEI claims cannot be verified based on evidence from Wikipedia alone; only the most relevant information is listed in the gold evidence set, making it difficult to extract a complete evidence set for NEI claims.