# OpenReview forum: "Enhancing Structured Evidence Extraction for Fact Verification"
_EMNLP/2023/Conference — EMNLP 2023 Main_

### Official Review · Reviewer_cEAj · 2023-08-02

**Soundness:** 4

**Excitement:**

4: Strong: This paper deepens the understanding of some phenomenon or lowers the barriers to an existing research direction.

**Paper Topic And Main Contributions:**

The authors are concerned with open-domain fact verification, specifically with the scenario where evidence to verify a claim has to be retrieved from a knowledge base consisting of both text and tables. The paper presents a method to extract of tables and associated cells by leveraging the row and column semantics of tables through a two-step pipeline: (i) the coarse grain extraction of tables from selected documents by computing a score over encodings of a table's rows and columns using a pre-trained tabular model, (ii) the fine-grained cell selection by building an evidence graph from the selected tables and separately extracted evidence sentences. The authors evaluate their system on FEVEROUS The authors show that their extraction module outperforms previous systems and when integrated into a full verification pipeline, they achieve new state-of-the-art performance on the FEVEROUS leaderboard.

**Questions For The Authors:**

- Wikipedia tables commonly also contain a table caption. Has the table caption been used? Was it useful at all to consider the caption for additional context?
- Whether a table cell is a header of the table or not is only considered for the TAPAS representation or also elsewhere (or maybe even nowhere)?
- Table 4 shows that removing either column or row semantics decreases performance only marginally. This is quite surprising to me, as I would expect that both semantics would be more complementary: i.e. for some tables, the semantics of the row is more important, for other tables the column (with a tendency towards the column being more important). Why is this not the case?
- I would have liked to see the standard error in the results to confirm the robustness of the approach. Can the experiments be rerun and mean + standard error be reported?
- Are the authors releasing their source code for reproducibility?

**Reasons To Accept:**

- The authors propose a simple and effective method for semi-structured evidence retrieval for fact-verification which improves over previous results substantially. While previous efforts have deployed both tabular and graph models for retrieval, the authors demonstrate that incorporating context beyond individual cells, but of entire rows and columns, is of essence. Tabular/Cell evidence retrieval is one of the key challenges of the FEVEROUS dataset and the authors provide a strong solution for this.
- Their approach is well motivated through selected examples. Figure 1 is well-chosen and demonstrates intuitively why a cell/row level extraction is beneficial.
-  The error analysis, broken down into challenging claims of different classes (e.g. Multi-hop Reasoning, Numerical Reasoning, etc.), highlights current gaps in evidence retrieval and motivates potential future directions.

**Reasons To Reject:**

- While the paper is clear on a high level, it is in part difficult to read and verbosely written. For instance, as mentioned above, Figure 1 showcases the advantages of a row/column-focused retrieval system due to the entire "Result" column being of relevance and the second-row providing additional context when considered in its entirety, yet the text does not concisely describe this.
- While the error analysis is interesting, parts of the analysis section are less of interest to me and appear less relevant to the story of the paper. The verdict classifier sensitivity regarding the evidence threshold seems a bit out of place since the paper does not propose the verdict prediction module itself. Similarly, the prediction performance for different veracity classes does not fit very well. The observations simply confirm previous work and the authors do not attempt to explain them (e.g. very low performance for NEI). Instead, I would wish the authors to focus more on the error analysis and provide qualitative examples of where their system fails.

**Reproducibility:**

4: Could mostly reproduce the results, but there may be some variation because of sample variance or minor variations in their interpretation of the protocol or method.

**Reviewer Confidence:**

4: Quite sure. I tried to check the important points carefully. It's unlikely, though conceivable, that I missed something that should affect my ratings.

**Typos Grammar Style And Presentation Improvements:**

- I strongly recommend the authors give the paper a proper pass for improvements in clarity and conciseness:
- Figure 3 is difficult to read. It is confusing that "Complete" is not an error class, but the other four ones are. This needs to be clarified in the figure and in the caption. Moreover, the numbers in 4.1 do not all match up with the Figure. l.448 mentions that the error rate for numerical reasoning is decreased to 10.3%, where is that in the figure? It shows 29.0% to me.
- l. 532 to l.534. I do not understand the connection of the sentence "The lexical matching approach [...] widely used in fact verification" to the previous one about question answering.
- l.544 and l.557 regarding Kotonya et al. (2021) are almost identical in content.
- Typos: l.56 construct, l.099 "most relevant rows and columns [for] the claim"
- Several sentences where necessary articles are omitted, resulting in ungrammatical sentences. I recommend passing through the paper with spell-checking software.
- Several lines with only a single word (e.g. l.110), should be avoided. Through slight rewording, the authors can be much more space efficient.

---

> ### Author Rebuttal · Authors · 2023-08-28
>
> We thank for your feedback on our paper. We appreciate your time and effort in reviewing our work. We have read all the questions for experiments and suggestions for writing. Below, we response to the concerns raised in the review.
>
> Q1: ...Has the table caption been used? ...
>
> A: As in the case we show in our paper, evidence extraction for tables is usually focused on specific cells which are only a small proportion of the table. And the table caption is more of a general of description of the table content. We haven't considered table captions yet. However, the use of table captions could be useful.
>
> Q2: ...table header only considered for TAPA representation?...
>
> A: Yes, the annotation of table headers is only used for TAPAS representation. A special token [H] is added in the cell to indicate it is a header. As for the evidence graph construction and cell selection, no specific manipulation for headers.
>
> Q3: ... Table 4 ...removing either column or row semantics decreases performance only marginally...
>
> A: I have to admit, this is surprising to me as well. I do not have solid explanations for this. I think the use of the whole table as input makes row or column representation still have context from other rows or columns. So the decrease is not as large as we expect.
>
> Q4: ...mean + standard error be reported?
>
> A: We have checked our logs and the mean value and standard error for evidence recall on dev set is 61.35 and 0.39(computed over 5 experiments). This describes the robustness of the crucial metrics. We would provide detailed standard errors for other metrics and experiments if the paper gets accepted
>
> Q5: code release:
>
> A: Sure, we will release our code along with the improved version of our paper soon.
>
> Q for Error Analysis in Reasons to Reject:
>
> A: We appreciate you pointing out the gap between the analysis of the verifier and our story. The verdict classifier sensitivity aims to explain why use the maximum number as the constraint for our evidence set (Please refer to Q2 for Reviewer #2 W4G6 for details). And we will provide more quantitative examples for error analysis of evidence extraction.
>
> I hope these answers address your concerns. We thank you again for your time and effort in reviewing our work. The suggestions for typos and presentation style are really helpful. We would definitely give the paper a proper pass for improvements.

---

### Official Review · Reviewer_W4G6 · 2023-08-05

**Soundness:** 4

**Excitement:**

3: Ambivalent: It has merits (e.g., it reports state-of-the-art results, the idea is nice), but there are key weaknesses (e.g., it describes incremental work), and it can significantly benefit from another round of revision. However, I won't object to accepting it if my co-reviewers champion it.

**Paper Topic And Main Contributions:**

This paper focuses on open-domain Fact Verification and proposed a framework that leverages row and column semantics of tables. The proposed method gains good results on benchmark dataset.

**Reasons To Accept:**

The method proposed in this paper is simple but effective. The ablation study and error analsysis help to better understand the advantages of the proposed method.

**Reasons To Reject:**

1. I may not fully understand the motivation behind the proposed method. What's the limitation of the existing method and how you address the issues? What's the advantages of your method and why? The paper can be improved if authors provide more insights in the introduction part.

2. From Table3, we can see that the proposed method underperforms the baseline methods on E-F1. Some explanations should be provided.

3. No code was provided. No code sharing promised.

**Reproducibility:**

2: Would be hard pressed to reproduce the results. The contribution depends on data that are simply not available outside the author's institution or consortium; not enough details are provided.

**Reviewer Confidence:**

4: Quite sure. I tried to check the important points carefully. It's unlikely, though conceivable, that I missed something that should affect my ratings.

---

> ### Author Rebuttal · Authors · 2023-08-28
>
> We thank for your feedback on our paper. We appreciate your time and effort in reviewing our work. We have read all the comments and suggestions. Below, we response to the concerns raised in the review.
>
> Q1: ...Motivation behind the proposed method...
>
> A: Please refer to Answer for Q2 for Review#1 CvBB. It explains the motivation and the advantages of our proposed method.
>
> Q2: ...the proposed method underperforms the baseline methods on E-F1...
>
> A: The idea is that for fact verification, the ultimate goal is the complete evidence set and accurate verdict prediction. Previous SOTA UnifEE uses a threshold to select evidence from top 5 sentences and top 25 cells which results in high F1. Such a method decreases recall of the complete evidence set. As shown in Figure 7, label accuracy remains unaffected by the change of evidence threshold (which affects precision, recall and F1). Therefore we choose to employ a maximum number of evidence as a constraint instead of the threshold to achieve optimal label accuracy and Feverous score at the final step.
>
> Q3: Code sharing
>
> A:  Sure, we will release our code along with the improved version of our paper soon.
>
> I hope this address your concerns. Thank you again for your time and consideration.

---

### Official Review · Reviewer_CvBB · 2023-08-05

**Soundness:** 3

**Excitement:**

3: Ambivalent: It has merits (e.g., it reports state-of-the-art results, the idea is nice), but there are key weaknesses (e.g., it describes incremental work), and it can significantly benefit from another round of revision. However, I won't object to accepting it if my co-reviewers champion it.

**Paper Topic And Main Contributions:**

This paper proposes a performant model for evidence extraction, the middle step of the three-step pipeline of open-domain fact verification. Unlike the previous evidence extraction methods, the proposed method first retrieves top tables based on the relevance score between the claim and the table rows or columns, then selects the relevant cells via a carefully designed graph neural network. The empirical results on the FEVEROUS benchmark demonstrate the effectiveness of the proposed method.

**Reasons To Accept:**

1. The paper is well-written and easy to follow.
2. The proposed method is simple yet effective.
3. The proposed method is well-ablated and analyzed.

**Reasons To Reject:**

1. The proposed method is very specialized, and it is unclear how it can be applied to tasks other than evidence extraction.
2. The performance success of the proposed method hasn't been fully explained in the context of the previous works, e.g., is the input semantic granularity difference the main reason that the proposed method outperforms baselines?

**Reproducibility:**

4: Could mostly reproduce the results, but there may be some variation because of sample variance or minor variations in their interpretation of the protocol or method.

**Reviewer Confidence:**

3: Pretty sure, but there's a chance I missed something. Although I have a good feel for this area in general, I did not carefully check the paper's details, e.g., the math, experimental design, or novelty.

---

> ### Author Rebuttal · Authors · 2023-08-28
>
> We thank you for your feedback on our paper. We appreciate your time and effort in reviewing our work. We have read all the comments and suggestions. Below, we respond to the raised.
>
> Q1: The proposed method is very specialized...
>
> A: Our work focuses on a challenging task that extracts evidence by combining texts and tables. This is a more realistic setting compared with other fact verification tasks focus on only one format of evidence. Our proposed method leverages row and column semantics to allow structured evidence to interact with unstructured evidence at a comparable semantic granularity. The same idea could be applied to open-domain question answering or information retrieval over other types of structured evidence.
>
> Q2:The performance success of the proposed method hasn't been fully explained...
>
> A: Previous works retrieve texts and tables separately without considering potential connection between evidence. The evidence graph allows interaction between texts and tables. *w/o evidence graph setting* in Table 5 indicates that the evidence graph is the main reason our method outperforms other baselines, especially for challenging claims that need multi-hop reasoning and combining tables and texts (shown in Figure 3).
> The proposed method also benefits a lot from leveraging row and column semantics. Rows and columns interact with sentences at comparable semantic granularity while still maintaining high-order semantics from tables. As shown in Figure 1, the *Result* column suggests *...lost all its games except for the one...* and redundant sentence and row evidence about *October 7 Match* corroborates each other.
>
> I hope this address your concerns. Thank you again for your time and consideration.

---

### Meta-Review · Area_Chair_3ZY9 · 2023-09-19

**Recommendation:** 4

**Metareview:**

Summary: The authors of the paper focus on verifying factual claims in open domains, specifically when evidence needs to be retrieved from a knowledge base consisting of text and tables. Their proposed approach involves extracting tables and the associated cells using a two-step pipeline. Firstly, they use a pre-trained tabular model to compute a score over encodings of a table's rows and columns to extract tables from selected documents. Secondly, they build an evidence graph using the selected tables and evidence sentences to select the cells. The authors demonstrate the effectiveness of their system by evaluating it on the FEVEROUS dataset.

Strengths: All the reviewers unanimously agree that this is a solid contribution with interesting implications. The approach used in this study is simple yet effective, yielding substantial improvements over previous models. The comprehensive experimentation conducted in this study greatly contributed to understanding the mechanism of the proposed method.

Weaknesses: I don't think there are any major weaknesses -- some of the weaknesses have been addressed during the discussion phase. The authors should take the feedback into account when preparing for the final version of the paper. For example, concerns about the baseline raised by Reviewer cEAj have not been addressed in full.

---

### Decision · Program_Chairs · 2023-10-07

**Decision:**

Accept-Main

**Comment:**

Summary: The authors of the paper focus on verifying factual claims in open domains, specifically when evidence needs to be retrieved from a knowledge base consisting of text and tables. Their proposed approach involves extracting tables and the associated cells using a two-step pipeline. Firstly, they use a pre-trained tabular model to compute a score over encodings of a table's rows and columns to extract tables from selected documents. Secondly, they build an evidence graph using the selected tables and evidence sentences to select the cells. The authors demonstrate the effectiveness of their system by evaluating it on the FEVEROUS dataset.

Strengths: All the reviewers unanimously agree that this is a solid contribution with interesting implications. The approach used in this study is simple yet effective, yielding substantial improvements over previous models. The comprehensive experimentation conducted in this study greatly contributed to understanding the mechanism of the proposed method.

Weaknesses: I don't think there are any major weaknesses -- some of the weaknesses have been addressed during the discussion phase. The authors should take the feedback into account when preparing for the final version of the paper. For example, concerns about the baseline raised by Reviewer cEAj have not been addressed in full.